# Simplification of a registry-based algorithm for ejection fraction prediction in heart failure patients: Applicability in cardiology centres of the Netherlands

**Elisa Dal Canto**[1,2]*, **Alicia Uijl**[2,3,4], **N. Charlotte Onland-Moret**[2], **Sophie H. Bots**[5], **Leonard Hofstra**[3,6], **Igor Tulevski**[3,6], **Folkert W. Asselbergs**[3], **Pim van der Harst**[7], **G. Aernout Somsen**[6], **Hester M. den Ruijter**[1]

1 Division Heart and Lungs, Laboratory of Experimental Cardiology, University Medical Center Utrecht, Utrecht, The Netherlands, 2 Julius Center for Health Sciences and Primary Care, University Medical Center Utrecht, Utrecht University, Utrecht, The Netherlands, 3 Department of Cardiology, Amsterdam University Medical Center, University of Amsterdam, Amsterdam, The Netherlands, 4 Division of Cardiology, Department of Medicine, Karolinska Institute, Stockholm, Sweden, 5 Division of Pharmacoepidemiology and Clinical Pharmacology, Utrecht Institute for Pharmaceutical Sciences, Utrecht University, Utrecht, The Netherlands, 6 Cardiology Centers of the Netherland, The Netherlands, 7 Division Heart and Lungs, Department of Cardiology, University Medical Center Utrecht, Utrecht, The Netherlands

* E.DalCanto@umcutrecht.nl

## Abstract

### Background

Left ventricular ejection fraction (EF) is used to categorize heart failure (HF) into phenotypes but this information is often missing in electronic health records or non-HF registries.

### Methods

We tested the applicability of a simplified version of a multivariable algorithm, that was developed on data of the Swedish Heart Failure Registry to predict EF in patients with HF. We used data from 4,868 patients with HF from the Cardiology Centers of the Netherlands database, an organization of 13 cardiac outpatient clinics that operate between the general practitioner and the hospital cardiologist. The algorithm included 17 demographical and clinical variables. We tested model discrimination, model performance and calculated model sensitivity, specificity, positive and negative predictive values for EF $\geq$ vs. <50% and EF $\geq$ vs. <40%. We additionally performed a multivariable multinomial analysis for all three separate HF phenotypes (with reduced, mildly reduced and preserved EF) HFrEF vs. HFmrEF vs. HFpEF. Finally, we internally validated the model by using temporal validation.

### Results

Mean age was 66 ±12 years, 44% of patients were women, 68% had HFpEF, 17% had HFrEF, and 15% had HFmrEF. The C-statistic was of 0.71 for EF $\geq$/< 50% (95% CI: 0.69–0.72) and of 0.74 (95% CI: 0.73–0.75) for EF $\geq$/< 40%. The model had the highest sensitivities for EF $\geq$50% (0.72, 95% CI: 0.63–0.75) and for EF $\geq$40% (0.70, 95% CI: 0.65–0.71).

**Data Availability Statement:** The CCN database are not publicly available and cannot be shared outside the University Medical Center Utrecht's

infrastructure due to ethical and data protection constraints. More specifically, the raw data contain potentially identifying and sensitive patient informationis and are kept by the data manager. The CCN is subject to the Dutch General Data Protection Regulation Implementation Act (Uitvoeringswet Algemene Verordening gegevensbescherming) which governs the processing of personal data. Data can however be made available upon reasonable request. Proposals for possible collaborations should be addressed to Dr Leonard Hofstra (L.Hofstra@cardiologiecentra. nl) or Prof Hester den Ruijter (H.M.denRuijter-2@umcutrecht.nl). Alternatively, research proposals or questions can be sent to the CCN Research and Innovation Manager Sebastiaan Blok (s.blok@cardiologiecentra.nl).

**Funding:** This study was supported by the Dutch Cardiovascular Alliance in the form of a RECONNEXT grant [2020B008] and by the Fondation Leducq in the form of a grant [16CVD02] to HMdR.

**Competing interests:** The authors have declared that no competing interests exist.

**Abbreviations:** HF, heart failure; EF, ejection fraction; HFpEF, Heart failure with preserved ejection fraction; HFrEF, Heart failure with reduced ejection fraction; HFmrEF, Heart failure with mildly reduced ejection fraction; EHRs, Electronic health records; CCN, Cardiology Centers of the Netherlands; GP, general practitioner; CVD, cardiovascular disease; eGFR, estimated glomerular filtration rate.

Similar results were achieved by the multinomial model, but the C-statistics for predicting HFpEF vs HFrEF was lower (0.61, 95% CI 0.58–0.63). The internal validation confirmed good discriminative ability.

## Conclusions

A simple algorithm based on routine clinical characteristics can help discern HF phenotypes in non-cardiology datasets and research settings such as research on primary care data, where measurements of EF is often not available.

## Introduction

Heart failure (HF) is a complex syndrome with high morbidity and mortality and its prevalence continues to rise [1]. Guidelines recommend the use of left ventricular ejection fraction (EF) assessed by echocardiography to categorize HF into three phenotypes: HF with preserved EF (HFpEF, EF≥50%), HF with mildly reduced EF (HFmrEF, EF = 40–49%) and HF with reduced EF (HFrEF, EF<40%) [2]. Because HF management differs according to the EF-based phenotype [2], accurate identification and categorization of patients is of utmost importance. HFpEF is currently the most common HF phenotype [3] and its diagnosis can be challenging [1]. Indeed HFpEF remains often undetected as symptoms and signs are not specific especially at rest, and affected subjects are therefore not correctly identified and promptly referred to specialized care. Moreover, treatment options are currently limited [4, 5].

Electronic health records (EHRs) comprise a wealth of information about patients diagnosis and treatment and they are widely used in research and clinical care, including HF care [6]. They have the potential to facilitate either HF research by for instance allowing wide screening for clinical trials and creation of registries, and might help to reduce variation in HF management thereby improving patient outcomes. However, EHRs sometimes lack imaging data and particularly information on EF is often not reported, thereby hampering ascertainment of the HF phenotype and identification of those with HFpEF.

A multivariable algorithm based on demographical and clinical characteristics has been developed using data of the Swedish Heart Failure Registry (SwedeHF) to predict EF in patients with HF [7]. The algorithm was externally validated in the CHECK-HF registry in The Netherlands, showing high discriminating power between HF phenotypes (C-statistic: 0.78 [95% confidence interval (CI) 0.77–0.78] for EF ≥50% and 0.76 (0.75–0.76) for EF ≥40%. Nevertheless, given the varying characteristics of HF populations, additional external validations would be essential to determine the algorithm generalizability and its ability to provide accurate predictions in distinct clinical context. Accordingly, our aim was to provide a further external validation of the HF algorithm in patients with chronic HF from the Cardiology Centers of the Netherlands (CCN) database. However, given the characteristics of our study population including the presence of missing data in some of the predictors, we derived and assessed the applicability of a simplified version, and thus more widely applicable, of the original model.

## Materials and methods

### Study population

The CCN is an organization of 13 cardiac outpatient clinics that operate between the general practitioner (GP) and the hospital [8]. Patients are referred to CCN on suspicion of

cardiovascular disease (CVD) by their GP and undergo an initial standardized diagnostic workup that includes echocardiography, ultrasound of the carotid arteries, exercise stress test, electrocardiography, laboratory tests and a consult with a nurse where anthropometrics and information on symptoms, medical history and medication use are collected. If needed, patients can be referred for further diagnostic workup to a hospital or invited to CCN for follow-up visits.

Among a total of 109,151 CCN patients, we selected and analyzed those diagnosed with HF between June 2007 and February 2018 (index dates), for whom a quantified EF was available at the time of diagnosis (S1 Fig). We defined HFrEF as EF <40%, HFmrEF as EF between 40% and 49%, and HFpEF as EF ≥50%. Patients missing information on demographical, anthropometrical and biochemical predictors included in the diagnostic algorithm were excluded if more than three months had passed between the assessment of EF and the assessment of such predictors (S1 Fig). A timeframe longer than three months from EF assessment was considered acceptable for information on co-morbidities and medication use. EF was assessed with biplane Simpson in 7% of cases and 93% of patients with Teich method.

The medical research ethics committee of the University Medical Center Utrecht approved this study (proposal number 17/359). Patient consent for publication was not required. The Cardiology Centers of the Netherlands data were made available under implied consent and transferred to the University Medical Center Utrecht under the Dutch Personal Data Protection Act. This study used data collected during the regular care process and did not subject participants to additional procedures or impose behavioral patterns on them. Finally, to the purpose of the present analysis, the CCN data were accessed between February 2022 and January 2024.

## Statistical analysis

Patients were stratified according to the HF phenotype and comparisons between the groups were carried out by analysis of variance for continuous variables and chi-squared for categorical variables. The main multivariable model included 22 variables: age, sex, NT-proBNP plasma level, New York Heart Association functional class, mean arterial pressure, heart rate, body mass index, estimated glomerular filtration rate (eGFR), history of ischemic heart disease, anemia, chronic obstructive pulmonary disease, diabetes, atrial fibrillation, hypertension, valvular heart disease, malignant cancer, device therapy, use of renin angiotensin system inhibitors (including ACE-inhibitors and angiotensin receptor blockers), beta-blockers, mineralocorticoid receptor antagonists, digoxin and diuretics. A simpler model excluded NT-proBNP and New York Heart Association class [7]. In the present analysis, we tested the applicability of a further simplified model which additionally excluded chronic obstructive pulmonary disease, malignant cancer and device therapy, as this information was unavailable in most patients. Missing values were present in a proportion ranging from 1.42% for the presence of hypertension to 39.17% for anemia (S1 Table) and imputed by using multiple imputation by chained equations (n = 10 imputations, *mice* package from R statistical software version 1.3.1093). Results were then pooled using Rubin's rule. We performed logistic regression to fit the model and used area under the receiver operating curves to discern model discrimination. Model discrimination was assessed for EF ≥ vs. <50% and EF ≥ vs. <40%. We assessed model performance with C-statistics. We calculated sensitivity, specificity, positive and negative predictive values along with 95% CI. Secondly, we used a multinomial logistic model to separately predict HFpEF, HFmrEF, and HFrEF (HFrEF was used as reference). In this case model C-statistics was calculated for all pairs of categories using the conditional risk method [9].

Finally, internal validation of the algorithm was carried out by using temporal validation. The dataset was split into training and testing sets based on the time of HF diagnosis, with 75% of the data used for training and the remaining 25% for testing, ensuring that the model was trained on historical data and evaluated on future observations. Logistic regression models were trained using the training set, and predictions were made on the testing set. Model accuracy was assessed using C-statistics and bootstrapping techniques were employed to estimate the 95% CIs. For the multinomial logistic model predicting HFpEF, HFmrEF, and HFrEF (with HFrEF as the reference), accuracy was calculated for all pairs of categories using the conditional risk method.

## Results

A total of 4,868 patients with HF were included (44% women, aged 66 ±12 years). Fifty-five percent of patients had hypertension, 15% had diabetes, and 20% had atrial fibrillation. Sixty-eight percent of patients had HFpEF, 17% had HFrEF, and 15% had HFmrEF (Table 1).

The performance of each predictor of the diagnostic algorithm to predict EF ≥50% vs. <50% and EF ≥40% vs. <40% is presented in Table 2. The strongest predictors for both EF ≥50% and EF ≥40% were female sex and presence of arterial hypertension. Use of diuretics, of mineralocorticoid receptor antagonists and presence of atrial fibrillation were the strongest predictors for EF <50% and EF <40%. Some predictors such as body mass index and advanced age, that were positively associated with EF ≥50/40% in the original study, did not show any significant associations in this analysis.

The model discriminated adequately for both EF ≥50% and EF ≥40% with a C-statistics of 0.71 (95% CI: 0.69–0.72) and of 0.74 (95% CI: 0.73–0.75) respectively (Figs 1 and 2). For EF <50% and EF <40% the model discriminately equally well with C-statistics of 0.71 (95% CI: 0.69–0.72) and of 0.74 (95% CI: 0.73–0.75) respectively (Figs 3 and 4).

The model had the highest sensitivities for EF ≥50% (0.72, 95% CI: 0.63–0.75) and for EF ≥40% (0.70, 95% CI: 0.65–0.71) (Table 3).

The results of the multinomial model are shown in Table 4. HFrEF was the reference category. Female sex and presence of arterial hypertension were the strongest predictors for HFmrEF. Predictors for HFpEF were the same as those for HFmrEF, but the associations were much stronger. C-statistics calculated for all pairs of outcome categories were similar to the logistic models for EF ≥50% or EF ≥40%, with C-statistics of 0.71 (95% 0.69–0.72) for HFmrEF vs HFrEF and of 0.74 (95% 0.72–0.76) for HFmrEF vs HFpEF. However, the discriminative performance for predicting HFpEF vs HFrEF was only moderate, with a C-statistic of 0.61 (95% CI 0.58–0.63).

Models were internally validated by temporal validation, showing good discriminative performance, with a C-statistics of 0.72 (95% CI: 0.71–0.73) for EF ≥/< 50% and of 0.69 (0.68–0.70) for EF ≥/< 40% (S2 Table).

## Discussion

In this study, we adapted a diagnostic algorithm originally designed for research purposes to predict EF among patients with HF in the Netherlands. This newly derived algorithm was applied to EHRs data obtained from Dutch cardiac screening centers, which represents a real-world clinical setting. Our findings demonstrated that the simplified version of the original algorithm performed adequately in predicting EF. As initially intended, this algorithm can be effectively utilized retrospectively on research data that have been collected on HF patients to ascertain the EF phenotype, thus serving as a valuable tool for further studies and investigations.

**Table 1. Baseline characteristics of the cardiology centers of the Netherlands population.**

| | Overall | HFrEF | HFmrEF | HFpEF | p[a] |
|---|---|---|---|---|---|
| **Demographics** | | | | | |
| **n** | 4868 | 827 | 719 | 3322 | |
| **Age (>75 vs <75 years)** | 65.8 (12.2) | 68.3 (11.8) | 67.4 (12.6) | 64.8 (12.1) | <0.001 |
| **Sex (female vs male)** | 2137 (43.9) | 266 (32.2) | 294 (40.9) | 1577 (47.5) | <0.001 |
| **Clinical variables** | | | | | |
| **MAP ≥ 90 mmHg, n (%)** | 917 (85.8) | 217 (76.4) | 171 (85.5) | 529 (90.4) | <0.001 |
| **Heart rate ≥ 70 bpm, n (%)** | 2605 (63.8) | 428 (74.0) | 347 (63.1) | 1830 (62.0) | <0.001 |
| **SBP, mmHg** | 147.5 (25.4) | 137.0 (23.9) | 148.3 (25.5) | 152.3 (24.7) | <0.001 |
| **DBP, mmHg** | 86.1 (14.8) | 84.8 (15.6) | 86.9 (16.6) | 86.4 (13.8) | 0.232 |
| **BMI, n (%)** | | | | | 0.003 |
| $< 18.5$ kg/m2 | 284 (26.4) | 90 (31.0) | 66 (32.2) | 128 (22.0) | |
| 18.5–24.9 kg/m2 | 9 (0.8) | 2 (0.7) | 1 (0.5) | 6 (1.0) | |
| 25–29.9 kg/m2 | 450 (41.8) | 117 (40.3) | 92 (44.9) | 241 (41.4) | |
| ≥ 30.0 kg/m2 | 334 (31.0) | 81 (27.9) | 46 (22.4) | 207 (35.6) | |
| **eGFR, n (%)** | | | | | |
| > = 90 mL/min/1.73m2 | 1438 (47.4) | 222 (40.1) | 177 (39.4) | 1039 (51.1) | <0.001 |
| 60–89.9 mL/min/1.73m2 | 1043 (34.4) | 182 (32.9) | 172 (38.3) | 689 (33.9) | |
| 59.9–30 mL/min/1.73m2 | 494 (16.3) | 125 (22.6) | 89 (19.8) | 280 (13.8) | |
| $<30$ mL/min/1.73m2 | 60 (2.0) | 24 (4.3) | 11 (2.4) | 25 (1.2) | |
| **Ischemic heart disease, n (%)** | 405 (8.3) | 75 (9.1) | 65 (9.0) | 265 (8.0) | 0.447 |
| **Anemia** | 263 (8.9) | 58 (11.7) | 56 (12.8) | 149 (7.4) | <0.001 |
| **Atrial fibrillation** | 969 (19.9) | 225 (27.2) | 181 (25.2) | 563 (16.9) | <0.001 |
| **Diabetes** | 741 (15.5) | 153 (19.1) | 115 (16.3) | 473 (14.4) | 0.003 |
| **Hypertension** | 2649 (55.2) | 326 (40.7) | 358 (50.7) | 1965 (59.7) | <0.001 |
| **Valvular disease** | 884 (18.2) | 192 (23.2) | 165 (22.9) | 527 (15.9) | <0.001 |
| **Therapy** | | | | | |
| **RAAS agents** | 3044 (64.2) | 576 (70.4) | 473 (66.8) | 1995 (62.1) | <0.001 |
| **Beta-blockers** | 2543 (53.7) | 540 (66.0) | 435 (61.4) | 1568 (48.8) | <0.001 |
| **Diuretics** | 2589 (54.6) | 582 (71.1) | 438 (61.9) | 1569 (48.8) | <0.001 |
| **MRA** | 755 (15.9) | 287 (35.1) | 141 (19.9) | 327 (10.2) | <0.001 |
| **Digoxin** | 340 (7.2) | 107 (13.1) | 70 (9.9) | 163 (5.1) | <0.001 |

[a] The p value refers to the comparisons between the groups, performed by analysis of variance (ANOVA) for continuous variables and chi-squared for trend for categorical variables.

Abbreviations: MAP: mean arterial pressure; BMI: body mass index; eGFR: estimated glomerular filtration rate; RAAS: renin-angiotensin-aldosterone system: MRA: mineralocorticoid receptor antagonists.

It's important to underscore the algorithm's primary domain of applicability: research-focused environments such as primary care data, non-cardiology research settings, and broader healthcare datasets where information about HF phenotypes is sporadic or unavailable. On the other hand, we realize that in clinical practice the use of echocardiography is essential and widely available for the diagnosis and management of HF [2, 10, 11]. However, our study focuses on research applications rather than replacing clinical assessments. While echocardiography remains the gold standard for EF assessment in clinical settings, our algorithm still provide a valuable research tool. Its integration into research initiatives has the potential to improve the accuracy of HF studies, especially in scenarios where EF measurements are lacking, such as EHRs. Furthermore, the utility of the algorithm might apply to

**Table 2. Multivariable logistic prediction models predicting EF ≥ 50% vs. EF < 50% and EF ≥ 40% vs. <40%.**

| Variables | LVEF ≥50% | | LVEF ≥40% | |
|---|---|---|---|---|
| | OR (95% CI) | P value | OR (95% CI) | P value |
| **Intercept** | 3.33 (1.83–7.08) | <0.001 | 6.30 (3.06–13.0) | <0.001 |
| **Age (>75 vs <75 years)** | 0.83 (0.68–1.01) | 0.060 | 0.85 (0.6–1.07) | 0.1287 |
| **Sex (female vs male)** | 1.81 (1.57–2.07) | <0.0001 | 2.08 (1.75–2.49) | <0.0001 |
| **MAP (≥ 90 vs < 90 mmHg)** | 1.00 (0.99–1.01) | 0.7632 | 1.00 (0.99–1.01) | 0.2979 |
| **Heart rate (≥ 70 vs < 70 bpm)** | 0.81 (0.70–0.94) | 0.0047 | 0.68 (0.57–0.82) | <0.0001 |
| **BMI** | | | | |
| < 18.5 vs 18.5–24.9 | 1.39 (0.44–4.37) | 0.5614 | 1.11 (0.31–4.02) | 0.8649 |
| 25–29.9 vs 18.5–24.9 | 1.07 (0.83–1.36) | 0.5825 | 1.14 (0.88–1.49) | 0.3095 |
| ≥ 30.0 vs 18.5–24.9 | 1.13 (0.89–1.43) | 0.2999 | 1.13 (0.83–1.54) | 0.4704 |
| **eGFR (mL/min/1.73m2)** | | | | |
| 60–89.9 vs > = 90 | 0.85 (0.72–1.01) | 0.0637 | 0.91 (0.73–1.13) | 0.4026 |
| 59.9–30 vs > = 90 | 0.75 (0.57–0.97) | 0.0298 | 0.73 (0.52–1.01) | 0.0598 |
| <30 vs > = 90 | 0.52 (0.30–0.90) | 0.0192 | 0.44 (0.25–0.78) | 0.0056 |
| **Ischemic heart disease** | 0.86 (0.68–1.08) | 0.1334 | 0.86 (0.65–1.14) | 0.3079 |
| **Anemia** | 0.92 (0.69–1.23) | 0.5767 | 1.10 (0.81–1.51) | 0.5244 |
| **Atrial fibrillation** | 0.65 (0.55–0.76) | <0.0001 | 0.66 (0.55–0.81) | <0.001 |
| **Diabetes** | 0.81 (0.67–0.97) | 0.0288 | 0.76 (0.61–0.96) | 0.0228 |
| **Hypertension** | 1.86 (1.62–2.14) | <0.0001 | 2.11 (1.78–2.51) | <0.0001 |
| **Valvular disease** | 0.74 (0.62–0.87) | <0.0001 | 0.82 (0.66–1.00) | 0.0503 |
| **RAAS agents** | 0.89 (0.76–1.04) | 0.1377 | 0.87 (0.72–1.05) | 0.1446 |
| **Beta-blockers** | 0.69 (0.60–0.79) | <0.0001 | 0.72 (0.60–0.86) | 0.0003 |
| **Diuretics** | 0.68 (0.58–0.79) | <0.001 | 0.66 (0.53–0.81) | <0.0001 |
| **MRA** | 0.41 (0.34–0.53) | <0.0001 | 0.37 (0.30–0.46) | <0.0001 |
| **Digoxin** | 0.81 (0.62–1.05) | 0.1048 | 0.83 (0.62–1.09) | 0.1863 |

MAP: mean arterial pressure; BMI: body mass index; eGFR: estimated glomerular filtration rate; RAAS: renin-angiotensin-aldosterone system: MRA: mineralocorticoid receptor antagonists.

other healthcare sources such as ICD-10 claims datasets which also often lack detailed clinical information.

This simplified version of the algorithm showed slightly worse performance compared to the performance demonstrated in the derivation and validation cohorts [7]. This might be due to the different characteristics of the study populations, with patients from the CCN database being younger, mostly affected by HFpEF, and healthier with lower prevalence of comorbidities such as diabetes and atrial fibrillation [8] compared to the SwedeHF and CHECK-HF. On the other hand, the aim of our study was to assess the algorithm's robustness in a population that differs in terms of age and health status from the derivation and validation cohorts. This enhanced our understanding of its generalizability and strengthened its potential for widespread use.

Additionally, we used fewer predictors than the original model in our analysis, and this has likely influenced our findings. Female sex and arterial hypertension were confirmed as strong predictors of HFpEF, while eGFR<30 mL/min/1.73m$^2$ and use of mineralocorticoid receptor antagonists were shown to be predictors of HFrEF. However, other predictors yielded different associations compared to those of the original study: atrial fibrillation predicted HFrEF rather than HFpEF, while some predictors such as body mass index and advanced age did not show significant associations.

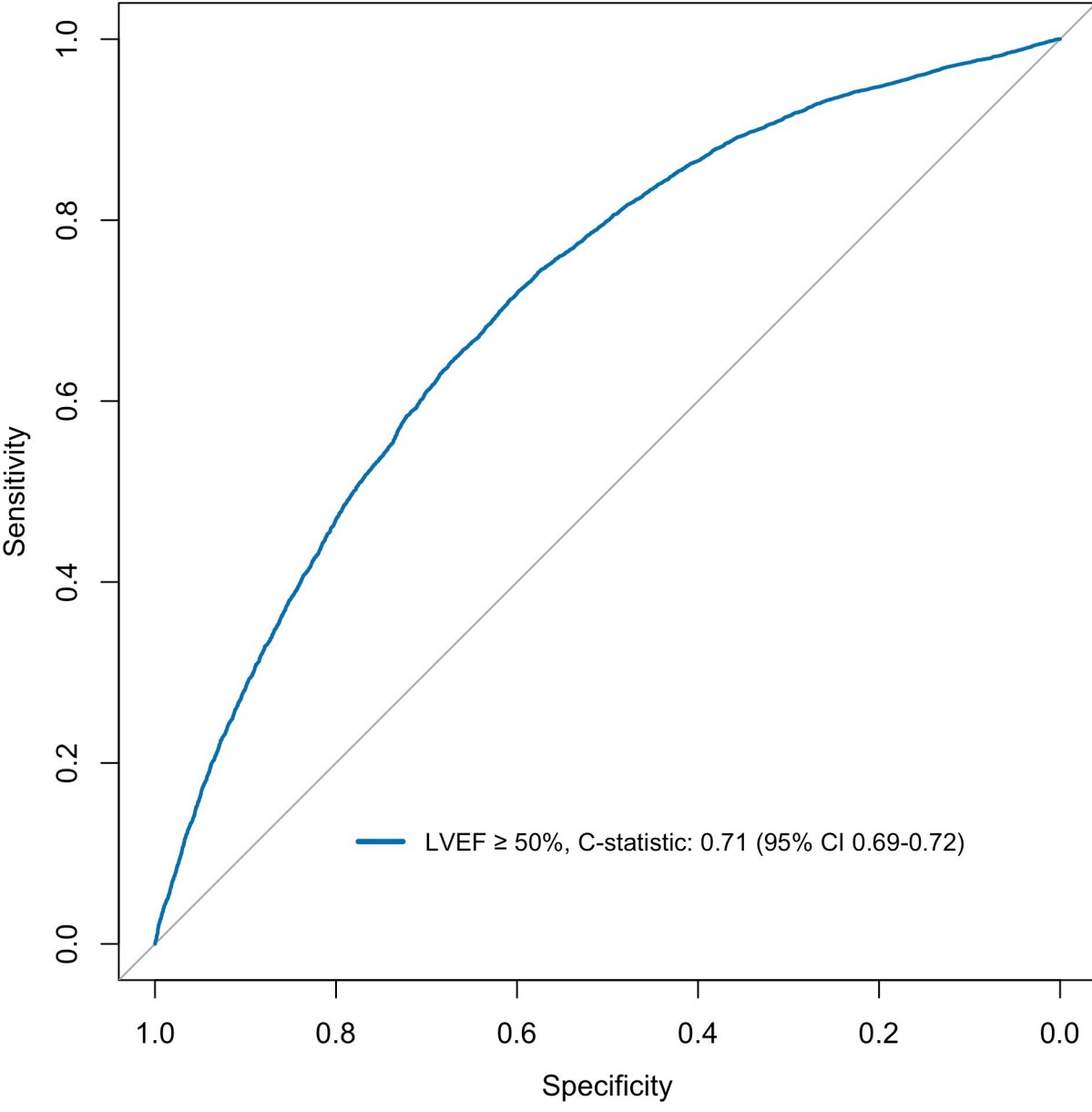

**Fig 1. Discrimination plot displaying ROC curve for logistic model EF cut-off ≥50%.**

The algorithm showed the highest sensitivities when used for the identification of patients with EF ≥50% and patients with EF ≥40%, of 0.72 and of 0.70 respectively. On the other hand, sensitivities for EF < 40% and EF<50% were not satisfactory. Finally, the associations with HFmrEF in the multinomial model were overall weak, probably because of a limited number of participants classified as HFmrEF in our study population. These findings and disparities are possibly due to changes in the distribution of patient characteristics observed when HFmrEF was combined with HFrEF and HFpEF, implying that the HFmrEF group should not be combined with neither of the other groups, being a separate and distinct clinical entity [12].

Previous studies have developed diagnostic algorithms to predict EF in HF populations [13, 14]. One of these algorithms was developed from Medicare claims and subsequently externally

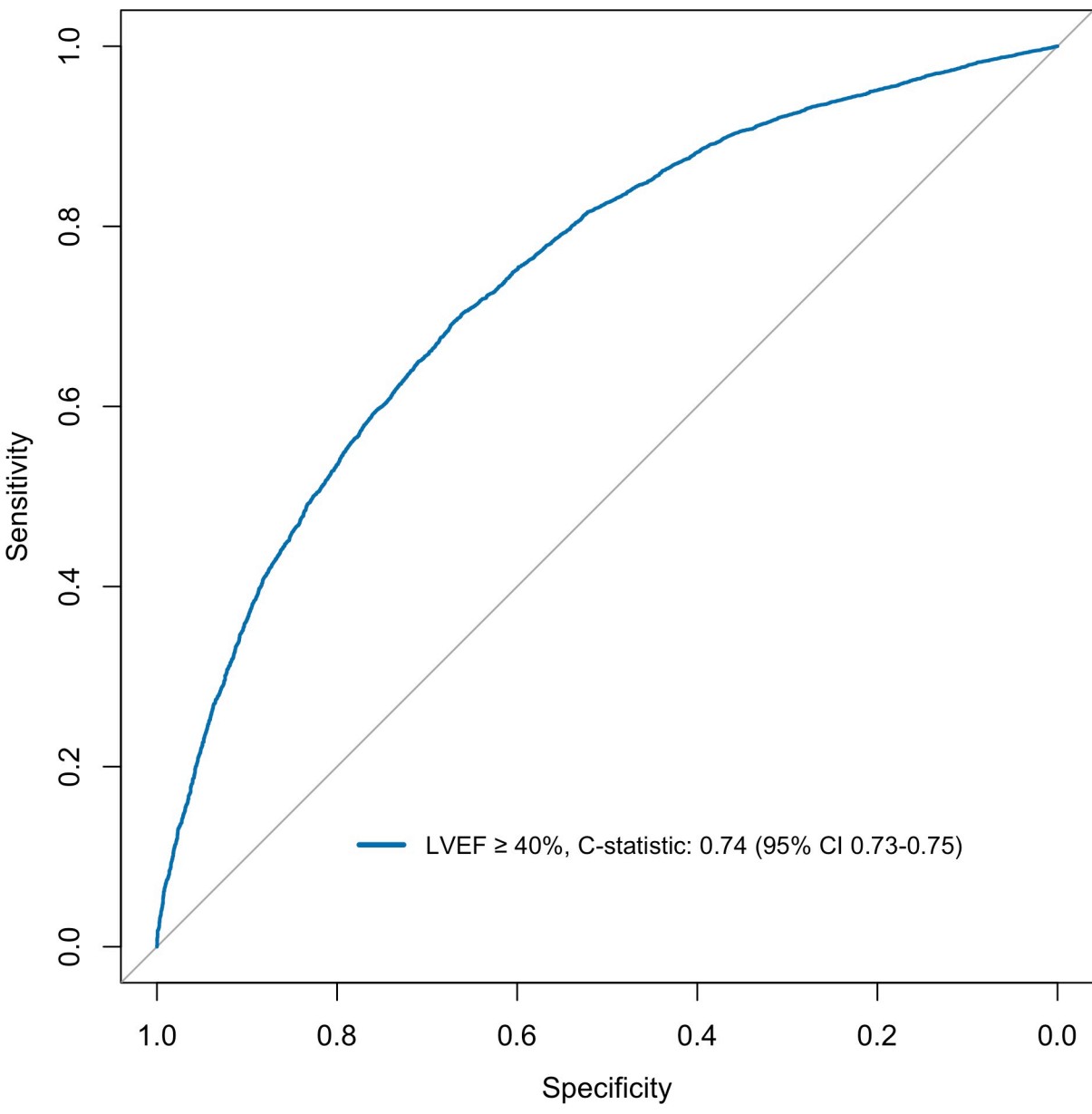

**Fig 2. Discrimination plot displaying ROC curve for logistic model EF cut-off ≥40%.**

validated in a sample of commercial insurance enrollees, demonstrating good accuracy [15]. However, this algorithm did not include laboratory test values and might therefore be more suitable in the context of insurance claims databases, in which this information is often missing.

## Study limitations

Some limitations of the present analysis should be acknowledged. Firstly, we had missing data on several predictors, which is a common issue in EHRs. Accordingly, we evaluated the performance of a model that included 17 out of the 22 original variables. It is important to note that this newly derived model was not tested neither validated in the original manuscript. However,

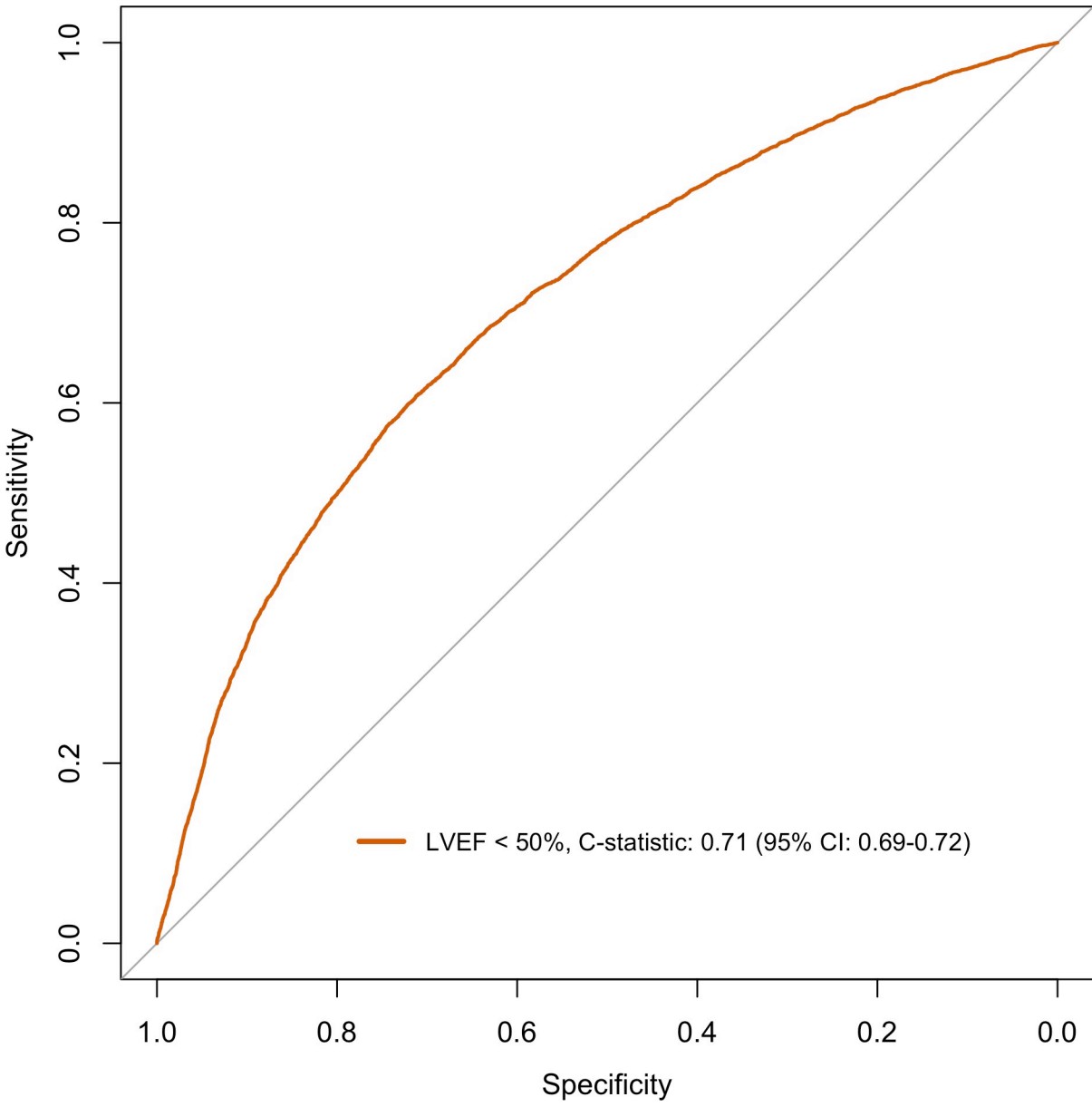

**Fig 3. Discrimination plot displaying ROC curve for logistic model EF cut-off <50%.**

this allowed us to demonstrate that a model containing fewer variables that are commonly available in clinical practice might be more widely applicable and enable to discriminate HF phenotypes, showing particular good performance in identifying HFpEF patients. Furthermore, we internally validated this model through temporal validation, which confirmed its good discriminative ability. Secondly, EF was often assessed only qualitatively and not quantitatively in the CCN database, which has resulted in a relatively small sample size. Among the cases with quantitatively assessed EF, in 93% of the cases the Teich method was used, which is less accurate than the biplane Simpson method, and tend to estimate the EF [10]. This might have resulted in the misclassification of some HFrEF patients as HFmrEF, potentially explaining the lower ability of the multinomial model to identify HFmrEF. Finally, for our analyses

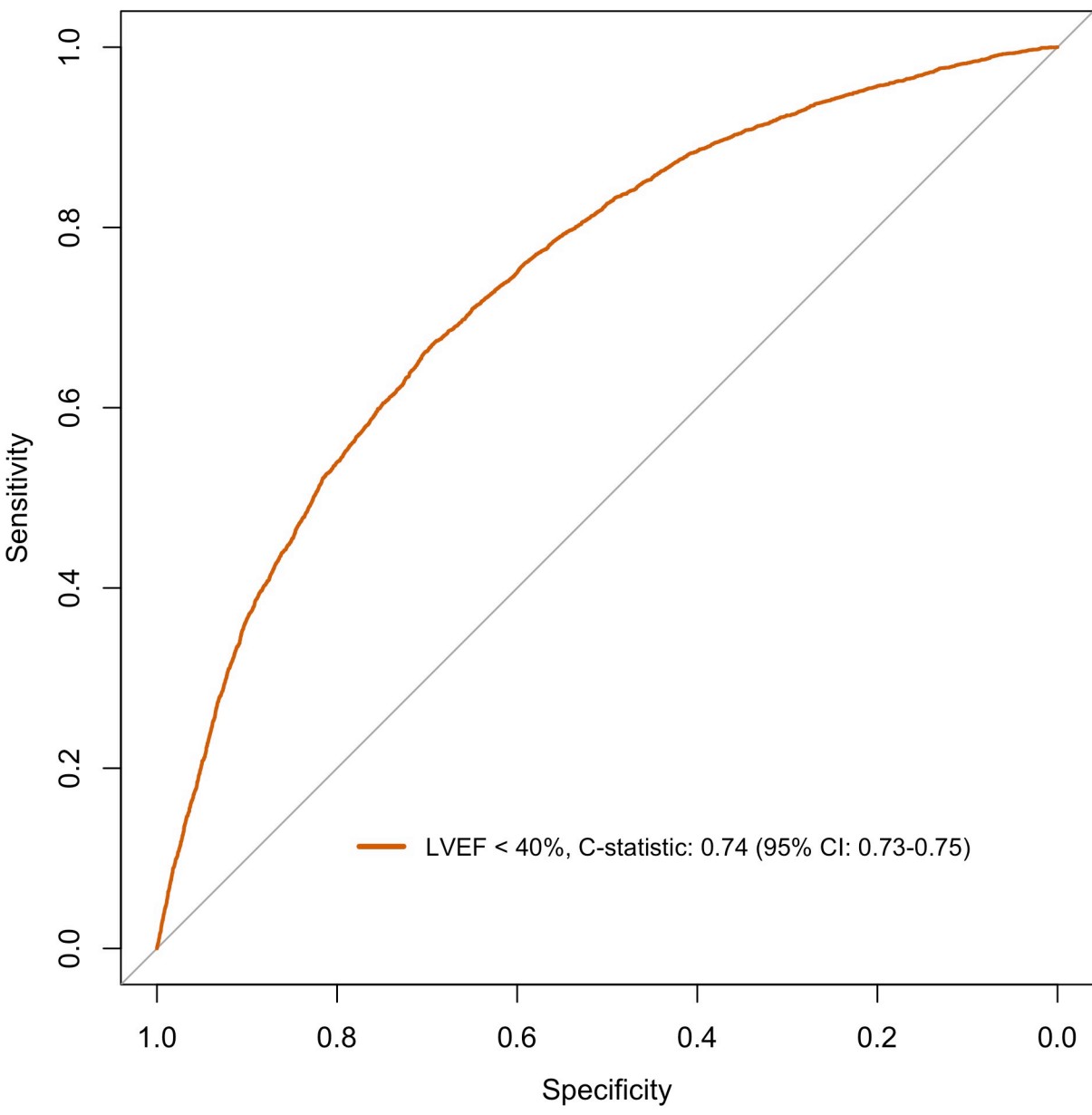

**Fig 4. Discrimination plot displaying ROC curve for logistic model EF cut-off <40%.**

we deemed a timeframe of less than three months between the determination of EF and the assessment of HF medications to be acceptable. We acknowledge that the dosage of these medications may be subject to adjustments over time as part of HF management, and

**Table 3. Sensitivity, specificity, positive and negative predictive values of the logistic prediction models.**

| | LVEF ≥50% | LVEF ≥40% | LVEF < 50% | LVEF < 40% |
|---|---|---|---|---|
| **Sensitivity** | 0.72 (0.63–0.75) | 0.70 (0.65–0.71) | 0.60 (0.57–0.68) | 0.67 (0.65–0.72) |
| **Specificity** | 0.60 (0.57–0.68) | 0.67 (0.65–0.72) | 0.72 (0.63–0.75) | 0.70 (0.65–0.71) |
| **Positive predictive value** | 0.80 (0.79–0.81) | 0.91 (0.90–0.92) | 0.50 (0.47–0.52) | 0.31 (0.29–0.32) |
| **Negative predictive value** | 0.50 (0.47–0.52) | 0.31 (0.29–0.32) | 0.80 (0.79–0.81) | 0.91 (0.90–0.92) |

**Table 4. Multinomial logistic prediction models predicting HFmrEF vs HFrEF and HFpEF vs HFrEF.**

| Variables | HFmrEF | | HFpEF | |
|---|---|---|---|---|
| | OR (95% CI) | P value | OR (95% CI) | P value |
| Intercept | 0.71 (0.27–1.83) | 0.4739 | 5.75 (2.72–12.1) | <0.001 |
| Age (>75 vs <75 years) | 0.96 (0.71–1.28) | 0.7822 | 0.81 (0.63–1.04) | 0.0978 |
| Sex (female vs male) | 1.56 (1.25–1.95) | <0.0001 | 2.27 (1.89–2.72) | <0.0001 |
| MAP ($\geq$ 90 vs < 90 mmHg) | 1.00 (0.99–1.01) | 0.2900 | 1.00 (0.99–1.01) | 0.3625 |
| Heart rate ($\geq$ 70 vs < 70 bpm) | 0.70 (0.55–0.89) | 0.0034 | 0.68 (0.56–0.82) | <0.0001 |
| BMI | | | | |
| < 18.5 vs 18.5–24.9 | 0.71 (0.10–4.57) | 0.7119 | 1.23 (0.32–4.70) | 0.7524 |
| 25–29.9 vs 18.5–24.9 | 1.14 (0.81–1.60) | 0.4359 | 1.14 (0.86–1.51) | 0.3414 |
| $\geq$ 30.0 vs 18.5–24.9 | 1.05 (0.70–1.55) | 0.8294 | 1.16 (0.85–1.58) | 0.3531 |
| eGFR (mL/min/1.73m2) | | | | |
| 60–89.9 vs > = 90 | 1.06 (0.81–1.37) | 0.6755 | 0.88 (0.70–1.10) | 0.2483 |
| 59.9–30 vs > = 90 | 0.85 (0.57–1.28) | 0.4441 | 0.69 (0.49–0.97) | 0.0354 |
| <30 vs > = 90 | 0.60 (0.25–1.28) | 0.1707 | 0.41 (0.22–0.74) | 0.0003 |
| Ischemic heart disease | 0.96 (0.67–1.37) | 0.8223 | 0.83 (0.63–1.12) | 0.2406 |
| Anemia | 1.27 (0.87–1.86) | 0.1767 | 1.03 (0.74–1.46) | 0.8212 |
| Atrial fibrillation | 0.89 (0.69–1.13) | 0.2958 | 0.61 (0.50–0.74) | <0.0001 |
| Diabetes | 0.83 (0.62–1.12) | 0.2385 | 0.74 (0.58–0.94) | 0.0151 |
| Hypertension | 1.54 (1.23–1.91) | 0.0001 | 2.31 (1.94–2.76) | <0.0001 |
| Valvular disease | 1.04 (0.81–1.34) | 0.7535 | 0.75 (0.61–0.93) | 0.0080 |
| RAAS agents | 0.92 (0.72–1.17) | 0.4911 | 0.85 (0.70–1.04) | 0.1146 |
| Beta-blockers | 0.95 (0.76–1.19) | 0.6654 | 0.67 (0.56–0.80) | <0.0001 |
| Diuretics | 0.85 (0.65–1.10) | 0.2077 | 0.62 (0.51–0.77) | <0.0001 |
| MRA | 0.51 (0.39–0.67) | <0.0001 | 0.33 (0.26–0.42) | <0.0001 |
| Digoxin | 0.93 (0.65–1.32) | 0.6801 | 0.78 (0.58–1.05) | 0.7738 |

MAP: mean arterial pressure; BMI: body mass index; eGFR: estimated glomerular filtration rate; RAAS: renin-angiotensin-aldosterone system: MRA: mineralocorticoid receptor antagonists.

discontinuation and initiation of these drugs might as well occur at notable rates. For instance, the EVOLUTION-HF study reported that 33–38% of patients discontinued ACE-inhibitors and angiotensin receptor blockers within 12 months [16]. However, these medications still represent the cornerstone of HF treatment and are typically prescribed and monitored closely in clinical practice, especially in the initial stages following diagnosis.

## Conclusions

Our investigation suggests that this simplified version of the algorithm shows promise in predicting EF, indicating its potential for retrospective utilization in research endeavors involving HF patients. By providing a means to characterize the EF phenotype, this algorithm could serve as a useful tool for guiding future studies within the HF domain, and particularly in the context of EHRs, where direct measurements of EF are not routinely available, such as in primary care settings.

## Supporting information

**S1 Table. Proportion of missing values (%) in each variable of the algorithm among the included patients.**
(DOCX)

**S2 Table. Internal validation of the models.**
(DOCX)

**S1 Fig. Flow chart of the study population selection.**
(TIF)

## Acknowledgments

The authors thank the other investigators, the staff, and the participants of the Cardiology Centers of the Netherlands for their valuable contributions.

## Author Contributions

**Conceptualization:** Elisa Dal Canto, Alicia Uijl, Sophie H. Bots, Igor Tulevski, G. Aernout Somsen, Hester M. den Ruijter.

**Data curation:** Alicia Uijl, Leonard Hofstra.

**Formal analysis:** Elisa Dal Canto.

**Funding acquisition:** Pim van der Harst, G. Aernout Somsen, Hester M. den Ruijter.

**Investigation:** Sophie H. Bots, Igor Tulevski, G. Aernout Somsen.

**Methodology:** N. Charlotte Onland-Moret.

**Project administration:** Sophie H. Bots, Leonard Hofstra, Igor Tulevski, Folkert W. Asselbergs, G. Aernout Somsen.

**Resources:** Leonard Hofstra, Folkert W. Asselbergs, Pim van der Harst, G. Aernout Somsen, Hester M. den Ruijter.

**Supervision:** N. Charlotte Onland-Moret, Pim van der Harst, Hester M. den Ruijter.

**Validation:** Elisa Dal Canto, Alicia Uijl.

**Visualization:** Elisa Dal Canto.

**Writing – original draft:** Elisa Dal Canto.

**Writing – review & editing:** Alicia Uijl, N. Charlotte Onland-Moret, Sophie H. Bots, Pim van der Harst, Hester M. den Ruijter.

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
