## [Decision Letter · Decision Letter 0]

19 Mar 2024

PONE-D-24-04912Simplification of a Registry-Based Algorithm for Ejection Fraction Prediction in Heart Failure Patients: Applicability in Cardiology Centers of the NetherlandsPLOS ONE

Dear Dr. Canto,

Thank you for submitting your manuscript to PLOS ONE. After careful consideration, we feel that it has merit but does not fully meet PLOS ONE’s publication criteria as it currently stands. Therefore, we invite you to submit a revised version of the manuscript that addresses the points raised during the review process.

We look forward to receiving your revised manuscript.

Kind regards,

Gianluigi Savarese

Academic Editor

PLOS ONE

4. In this instance it seems there may be acceptable restrictions in place that prevent the public sharing of your minimal data. However, in line with our goal of ensuring long-term data availability to all interested researchers, PLOS’ Data Policy states that authors cannot be the sole named individuals responsible for ensuring data access (http://journals.plos.org/plosone/s/data-availability#loc-acceptable-data-sharing-methods).

5. Please remove your figures from within your manuscript file, leaving only the individual TIFF/EPS image files, uploaded separately. These will be automatically included in the reviewers’ PDF.

Reviewers' comments:

Reviewer's Responses to Questions

**Comments to the Author**

1. Is the manuscript technically sound, and do the data support the conclusions?

Reviewer #1: Yes

Reviewer #2: Partly

2. Has the statistical analysis been performed appropriately and rigorously? 

Reviewer #1: Yes

Reviewer #2: Yes

3. Have the authors made all data underlying the findings in their manuscript fully available?

Reviewer #1: Yes

Reviewer #2: No

4. Is the manuscript presented in an intelligible fashion and written in standard English?

Reviewer #1: Yes

Reviewer #2: Yes

5. Review Comments to the Author

Reviewer #1: In this manuscript, Dr. Del Canto and colleagues evaluated the generalizability of an algorithm for ejection fraction prediction based on the Swedish Heart Failure Registry on a cohort of patients from the Cardiology Centers of the Netherlands database.

Overall, the article is nicely written, but some issues need to be addressed by the authors:

- Page 4: Considering the high proportion of patients with Teich method as EF assessment, the authors may perform a sensitivity analysis only in patients with EF assessed via Simpson. Even If in the original publication (Reference 7) the method for EF assessment is not reported, a high difference is present between the Teich and Simpson methods in evaluating EF (PMID: 25712077).

- Page 5: While I understand that many missing data and difference between registry may lead to different models, the authors are evaluating a model with 17 variables from the original 22 variables model, which was not tested and/or validated in the original manuscript. This should be clearly stated in the limitation section, or the authors may test this simplified model in the original cohorts for validation.

- Page 5: I would suggest the authors to use 10 imputations in mice instead of 5 and to report in a supplementary table the percentage of missing data for the evaluated variables.

- Page 6: The authors may consider just reporting the model values on sensitivity/specificity etc. in a separate table

- Page 8: I would remove this section: “Its integration into research initiatives holds the promise of enhancing the comprehensiveness and accuracy of HF studies, especially in scenarios where EF measurements are lacking. While the algorithm's complexity might make it better suited for research applications, its insights can indirectly impact clinical care. By contributing to a deeper understanding of HF phenotypes and their associations with various clinical variables, the algorithm indirectly enriches the broader clinical knowledge base. This can, in turn, inform the development of simpler clinical tools that are tailored for quick assessment in various healthcare settings.”

- Page 10: Please tone down the conclusion section

- Why are the figures reported two times?

- Table 1: The reported p-value is from what test? Do RAAS agents include ARNi?

-Figures 1a and 1b: Please change “>=” with “≥”.

- Please also note that the text requires careful proofreading since there are several grammar lapses.

Reviewer #2: Dal Canto et al evaluated a simple model aiming to identify ejection fraction categories from other variables. Such a model, if well-performing, would be valuable for research purposes since EF is often lacking in ICD-code data and EHR. The model the authors evaluated was a simplified version of a model that was previously developed in SwedeHF and validated in CHECK-HF (doi: 10.1002/ehf2.12779). When applying a simplified version of this algorithm in the Cardiology Centers of Netherlands (CCN) database, the corresponding C-indices were 0.71 and 0.74, respectively. The manuscript is overall well-written. I have the following comments for the authors.

- Throughout the manuscript, the wording can give the impression that this was an external validation of the model developed in SwedeHF. E.g. page 3, background “[…] additional external validations are essential […]” and page 7, discussion “[…] allowing us to validate its performance […]”. Although the variables were selected from a previously validated model, this was not a validation of that original model, but rather the derivation of a new model and no validation. This could be clarified throughout the manuscript.

- In context of comment 1, did the authors perform/consider doing any validation procedures, e.g. temporal validation?

- The background/discussion centres on the relevance of a EF-predicting model for EHR data – the authors might consider expanding also on the utility in ICD-10 / claims databases which also typically lack EF.

- Patients were excluded based on several parameters, including missingness of data. I suggest to include a patient selection flowchart.

- Missing data: I suggest to provide details on missing data on a per-variable level. Also, with 38% missing for anemia (which was included in the model) it is likely that 5 imputations is too few.

- “The associations with HFmrEF were overall weak, probably because of a limited number of participants with HFmrEF in our study population”. This should be moved to discussion.

6. PLOS authors have the option to publish the peer review history of their article (what does this mean?). If published, this will include your full peer review and any attached files.

Reviewer #1: No

Reviewer #2: No

---

## [Author Response · Author response to Decision Letter 0]

13 May 2024

Submission ID: PONE-D-24-04912

Manuscript title: Simplification of a Registry-Based Algorithm for Ejection Fraction Prediction in Heart Failure Patients: Applicability in Cardiology Centers of the Netherlands

Response to editor and reviewers 

We would like to thank the reviewers and editor very much for carefully reviewing our manuscript. Their review helped us to improve the manuscript. We have addressed all comments below, point-by-point. In addition, we have revised the manuscript (changes are marked in blue text).

Reviewer 1

In this manuscript, Dr. Dal Canto and colleagues evaluated the generalizability of an algorithm for ejection fraction prediction based on the Swedish Heart Failure Registry on a cohort of patients from the Cardiology Centers of the Netherlands database. Overall, the article is nicely written, but some issues need to be addressed by the authors:

1. Page 4: Considering the high proportion of patients with Teich method as EF assessment, the authors may perform a sensitivity analysis only in patients with EF assessed via Simpson. Even If in the original publication (Reference 7) the method for EF assessment is not reported, a high difference is present between the Teich and Simpson methods in evaluating EF (PMID: 25712077).

Reply: We appreciate the reviewer’s suggestion to perform a sensitivity analysis focusing on patients with EF assessed via the Simpson method. However, due to the small proportion of patients (only 341) with EF measured using this method, conducting a meaningful sensitivity analysis is not feasible. Nevertheless, we acknowledge the potential impact of using different EF assessment methods on the classification of heart failure phenotypes. The use of the Teich method, which often overestimates EF, may have led to some misclassification, particularly in distinguishing between HFmrEF and other phenotypes. This might help to explain why the association with HFmrEF in the multinomial model were overall weaker. We have revised the discussion section to acknowledge this limitation and its potential implications for our analysis: “Among the cases with quantitatively assessed EF, in 93% of the cases the Teich method was used, which is less accurate than the biplane Simpson method, and tend to estimate the EF. This might have resulted in the misclassification of some HFrEF patients as HFmrEF, potentially explaining the lower ability of the multinomial model to identify HFmrEF. (page 10).”

2. Page 5: While I understand that many missing data and difference between registry may lead to different models, the authors are evaluating a model with 17 variables from the original 22 variables model, which was not tested and/or validated in the original manuscript. This should be clearly stated in the limitation section, or the authors may test this simplified model in the original cohorts for validation.

Reply: thanks for this comment, indeed this represents an important limitation of our analysis. We now stated this more clearly in the limitations section as: “Firstly, we had missing data on several predictors, which is a common issue in EHRs. Accordingly, we evaluated the performance of a model that included 17 out of the 22 original variables. It is important to note that this newly derived model was not tested neither validated in the original manuscript.” Furthermore, in response to comment number 1 from reviewer 2, we revised introduction and discussion to point out that the model derived and tested in our study indeed represents a new model which differs from the original one. By doing so we made it clear that our study does not represent an external validation of the original model. However, in this revised version of the paper we have conducted internal validation of the model through temporal validation, that confirmed its good discriminative ability. We believe that this procedure has improved the robustness of our findings.

3. Page 5: I would suggest the authors to use 10 imputations in mice instead of 5 and to report in a supplementary table the percentage of missing data for the evaluated variables.

Reply: Indeed we agree on the fact that having ten imputation might be more suitable to our data, and revised our approach accordingly, by incrementing the number of imputations. The results that we present in this revised version are based on the new analysis and on the newly imputed dataset. There were some minor changes; in particular, eGFR <30 mL/min/1.73mq is no longer one of the strongest predictor of EF<50% and 40%, while use of diuretics, of MRA and presence of atrial fibrillation are now the variables more strongly associated to EF<40% and EF<50%. There were no changes in the predictors of EF≥50% and ≥40%. For the multinomial model, age > 75 years old is no longer significantly associated to HFpEF. We revised the tables and the results section accordingly: “The strongest predictors for both EF ≥50% and EF ≥40% were female sex and presence of arterial hypertension. Use of diuretics, of mineralocorticoid receptor antagonists and presence of atrial fibrillation were the strongest predictors for EF <50% and EF <40%. (page 7)” 

4. Page 6: The authors may consider just reporting the model values on sensitivity/specificity etc. in a separate table.

Reply: we agree on this and added the model sensitivity, specificity, positive and negative predictive values in a table, which now is Table 3. We kept only one sentence in the text to report the highest sensitivities: “The model had the highest sensitivities for ejection fraction ≥50% (0.70, 95% CI: 0.63–0.71) and for ejection fraction ≥40% (0.63, 95% CI: 0.61–0.75) (page 7).”

5. Page 8: I would remove this section: “Its integration into research initiatives holds the promise of enhancing the comprehensiveness and accuracy of HF studies, especially in scenarios where EF measurements are lacking. While the algorithm's complexity might make it better suited for research applications, its insights can indirectly impact clinical care. By contributing to a deeper understanding of HF phenotypes and their associations with various clinical variables, the algorithm indirectly enriches the broader clinical knowledge base. This can, in turn, inform the development of simpler clinical tools that are tailored for quick assessment in various healthcare settings.”

Reply: We left this section out and replaced with a single more concise sentence. Moreover, this section was further revised to address comment number 3 of reviewer 2 on the potential utility of the algorithm in ICD-10 claims datasets (page 8). “Its integration into research initiatives has the potential to improve the accuracy of HF studies, especially in scenarios where EF measurements are lacking, such as EHRs. Furthermore, the utility of the algorithm might apply to other healthcare sources such as ICD-10 claims datasets which also often lack detailed clinical information.”

6. Page 10: Please tone down the conclusion section

Reply: thank you for your feedback, we have revised this section to convey the findings of our study in a more balanced and measured manner, aligning with the overall tone of the manuscript: “Our investigation suggests that this simplified version of the algorithm shows promise in predicting EF, indicating its potential for retrospective utilization in research endeavors involving HF patients. By providing a means to characterize the EF phenotype, this algorithm could serve as a useful tool for guiding future studies within the HF domain, and particularly in the context of EHRs, where direct measurements of EF are not routinely available, such as in primary care, or other non-cardiology health settings such as GP settings.”

7. Why are the figures reported two times?

Reply: we removed one recurrence of the figures. 

8. Table 1: The reported p-value is from what test? Do RAAS agents include ARNi?

Reply: The reported p value in table 1 refers to the comparisons between the heart failure phenotypes groups, which was performed by analysis of variance (ANOVA) for continuous variables and chi-squared for trend for categorical variables. We reported this information now as a footnote to the table and in the methods section. RAAS agents include ACE-inhibitors and angiotensin receptors blockers but not angiotensin receptor-neprilysin inhibitors (ARNi). That is because we created a variable that reflects the original RAAS agents variable of the algorithm, which did not include ARNi as well. We clarified this in the text on page 5. 

9. Figures 1a and 1b: Please change “>=” with “≥”.

Reply: we revised the figures accordingly. 

10. Please also note that the text requires careful proofreading since there are several grammar lapses.

Reply: thanks for this feedback. We carefully reviewed and proofread the manuscript to correct any grammar mistakes. 

Reviewer 2

Dal Canto et al evaluated a simple model aiming to identify ejection fraction categories from other variables. Such a model, if well-performing, would be valuable for research purposes since EF is often lacking in ICD-code data and EHR. The model the authors evaluated was a simplified version of a model that was previously developed in SwedeHF and validated in CHECK-HF (doi: 10.1002/ehf2.12779). When applying a simplified version of this algorithm in the Cardiology Centers of Netherlands (CCN) database, the corresponding C-indices were 0.71 and 0.74, respectively. The manuscript is overall well-written. I have the following comments for the authors.

1. Throughout the manuscript, the wording can give the impression that this was an external validation of the model developed in SwedeHF. E.g. page 3, background “[…] additional external validations are essential […]” and page 7, discussion “[…] allowing us to validate its performance […]”. Although the variables were selected from a previously validated model, this was not a validation of that original model, but rather the derivation of a new model and no validation. This could be clarified throughout the manuscript.

Reply: We appreciate the reviewer's feedback and have revised the relevant sections of the manuscript to address this concern. Our original aim was to provide a further external validation of the original model. However, given the issue of missing data in some of the predictors, we derived and tested a new and simplified model, which included 17 out of the 22 original variables. In our revised version we clarified this in both the introduction as: “Accordingly, our aim was to provide a further external validation of the HF algorithm in patients with chronic HF from the Cardiology Centers of the Netherlands (CCN) database. However, given the characteristics of our study population including the presence of missing data in some of the predictors, we derived and assessed the applicability of a simplified version, and thus more widely applicable, of the original model” (page 4). And in the discussion as: “In this study, we adapted a diagnostic algorithm originally designed for research purposes to predict EF among patients with HF in the Netherlands. This newly derived algorithm was applied to EHRs data obtained from Dutch cardiac screening centers, which represents a real-world clinical setting. Our findings demonstrated that the simplified version of the original algorithm performed adequately in predicting EF.” Furthermore, in response to comment number 2 of reviewer 1, we revised the limitation section to acknowledge the fact that this new model we derived was not validated in the original manuscript (page 10): “Firstly, we had missing data on several predictors, which is a common issue in EHRs. Accordingly, we evaluated the performance of a model that included 17 out of the 22 original variables. It is important to note that this newly derived model was not tested neither validated in the original manuscript.

2. In context of comment 1, did the authors perform/consider doing any validation procedures, e.g. temporal validation?

Reply: After careful consideration of the reviewer's feedback, we performed temporal validation to assess the performance of the derived model. This involved splitting the dataset into training and testing sets based on time, with 75% of the data used for training the model and 25% for testing. Logistic and multinomial regression models were trained using the training set, and predictions were made on the testing set to evaluate model accuracy. This is carefully explained in the methods section. Results are presented in the results section as well as in Supplementary Table 2. Validation showed good discriminative performance of the model with a C-statistics of 0.72 (95% CI: 0.71 – 0.73) for EF ≥ 50% and of 0.69 (0.68 – 0.70) for EF ≥40%. We would like to thank the reviewer for this comment as we believe that the inclusion of temporal validation enhances the robustness and generalizability of our findings.

3. The background/discussion centers on the relevance of a EF-predicting model for EHR data – the authors might consider expanding also on the utility in ICD-10 / claims databases which also typically lack EF.

Reply: thanks for this comment. We agree that ICD-10 / claims databases might represent an additional application for the algorithm. We now have acknowledged this in the discussion section as: “Its integration into research initiatives has the potential to improve the accuracy of HF studies, especially in scenarios where EF measurements are lacking, such as EHRs. Furthermore, the utility of the algorithm might apply to other healthcare sources such as ICD-10 claims datasets which also often lack detailed clinical information.” (page 9). 

4. Patients were excluded based on several parameters, including missingness of data. I suggest to include a patient selection flowchart.

Reply: in this revised version we have included a Flow chart of the Study Population selection (S1 Figure), which clarifies the steps that have been taken in this regard. 

5. Missing data: I suggest to provide details on missing data on a per-variable level. Also, with 38% missing for anemia (which was included in the model) it is likely that 5 imputations is too few.

Reply: We provided the information on the amount of missing values for each variable of the algorithm in S1 Table (Supporting information). Furthermore, we incremented the number of imputations to 10, also in response to comment number 3 of reviewer 1. 

6. “The associations with HFmrEF were overall weak, probably because of a limited number of participants with HFmrEF in our study population”. This should be moved to discussion.

Reply: we moved this statement to the discussion (page 9), which has been carefully revised in response to several comments from both reviewers.

---

## [Decision Letter · Decision Letter 1]

30 Jul 2024

PONE-D-24-04912R1Simplification of a Registry-Based Algorithm for Ejection Fraction Prediction in Heart Failure Patients: Applicability in Cardiology Centres of the NetherlandsPLOS ONE

Dear Dr. Canto,

Thank you for submitting your manuscript to PLOS ONE. After careful consideration, we feel that it has merit but does not fully meet PLOS ONE’s publication criteria as it currently stands. Therefore, we invite you to submit a revised version of the manuscript that addresses the points raised during the review process.

We look forward to receiving your revised manuscript.

Kind regards,

Satoshi Higuchi

Academic Editor

PLOS ONE

Reviewers' comments:

Reviewer's Responses to Questions

**Comments to the Author**

1. If the authors have adequately addressed your comments raised in a previous round of review and you feel that this manuscript is now acceptable for publication, you may indicate that here to bypass the “Comments to the Author” section, enter your conflict of interest statement in the “Confidential to Editor” section, and submit your "Accept" recommendation.

Reviewer #1: All comments have been addressed

Reviewer #2: (No Response)

2. Is the manuscript technically sound, and do the data support the conclusions?

Reviewer #1: Yes

Reviewer #2: Yes

3. Has the statistical analysis been performed appropriately and rigorously? 

Reviewer #1: Yes

Reviewer #2: No

4. Have the authors made all data underlying the findings in their manuscript fully available?

Reviewer #1: Yes

Reviewer #2: Yes

5. Is the manuscript presented in an intelligible fashion and written in standard English?

Reviewer #1: Yes

Reviewer #2: Yes

6. Review Comments to the Author

**Reviewer #1:** I thank the authors for replying and integrating my comments.

As it is I have no further comments for the authors.

**Reviewer #2:** I thank the authors for addressing the previous comments. I have the following comments remaining:

1. The numbers in Table 3 seem inconsistent. For example, since LVEF >=50% has 0.70 sensitivity, shouldn't I expect LVEF <50% to have 0.7 specificity? (since TP and FN for LVEF>=50% should be the same as TN and FP for LVEF<50%). Rather, it seems the LVEF<50% column contains the FNR (1-specificity) instead of sensitivity and the FPR (1-sensitivity) instead of specificity. There are similar apparent inconsistencies for PPV and NPV in Table 3 and also for the C-statistics in Supplementary Table S2.

2. The omission of ARNi from the model strikes me as strange. Of the drugs that could be included in the RASi-variable (ACEi, ARB, and ARNi), ARNi is likely the far best at distinguishing between the EF phenotypes. Moreover, ARNi technically includes an ARB. How much this omission affects model performance is difficult to speculate without knowing the time period of enrollment and how many patients used ARNi. I highly suggest to consider including it in the RASi variable, as well as to disclose ACEi/ARB and ARNi separately in the characteristics table.

3. Please clarify in the methods between which dates these patients had their index dates. It says CCN data were accessed between Feb 2022 and Jan 2024, but I interpret this literally as data access and not when the patients presented in the clinic.

4. Page 11: "While we acknowledge that the dosage of these medications may be subject to adjustments over time as part of HF management, it is unlikely that the discontinuation or initiation of these drugs would occur significantly long after the initial diagnosis of HF. This is because these HF medications represent the cornerstone of HF treatment and are typically prescribed and monitored closely in clinical practice." This sentence seems at odds with observations from several health care systems. E.g. in EVOLUTION-HF (10.1016/j.jchf.2022.08.009) 33-38% of patients discontinued ACEi/ARB within 12 months.

5. I suggest to change the term "mid-range" EF to "mildly reduced" EF to align with currently accepted terminology.

6. Conclusion: Just a minor comment but was the distinction between primary care and GP settings made on purpose? "[...] where direct measurements of EF are not routinely available, such as in primary care, or other non-cardiology health settings such as GP settings."

7. PLOS authors have the option to publish the peer review history of their article (what does this mean?). If published, this will include your full peer review and any attached files.

Reviewer #1: No

Reviewer #2: No

---

## [Author Response · Author response to Decision Letter 1]

2 Aug 2024

Submission ID: PONE-D-24-04912R1

Manuscript title: Simplification of a Registry-Based Algorithm for Ejection Fraction Prediction in Heart Failure Patients: Applicability in Cardiology Centers of the Netherlands

Response to editor and reviewers 

We would like to thank the reviewers and editor very much for carefully reviewing our manuscript. We appreciate their detailed review and the opportunity to improve our manuscript. We have addressed all comments below, point-by-point. In addition, we have revised the manuscript (changes are marked in blue text).

Reviewer #1: I thank the authors for replying and integrating my comments.

As it is I have no further comments for the authors.

Reply: We are pleased to know that our rebuttal addressed the reviewer’s concerns satisfactorily. Thank you for your thorough review and valuable feedback on our manuscript. 

Reviewer #2: I thank the authors for addressing the previous comments. I have the following comments remaining:

1. The numbers in Table 3 seem inconsistent. For example, since LVEF >=50% has 0.70 sensitivity, shouldn't I expect LVEF <50% to have 0.7 specificity? (since TP and FN for LVEF>=50% should be the same as TN and FP for LVEF<50%). Rather, it seems the LVEF<50% column contains the FNR (1-specificity) instead of sensitivity and the FPR (1-sensitivity) instead of specificity. There are similar apparent inconsistencies for PPV and NPV in Table 3 and also for the C-statistics in Supplementary Table S2.

Reply: Thank you for your valuable feedback. We have carefully reviewed your comment regarding the inconsistencies in Table 3 and the C-statistics in Supplementary Table S2. We acknowledge the importance of accurately presenting these metrics, and apologize for the errors in reporting these results. Upon re-evaluation of our calculations and the analysis of both LVEF >= 50% and LVEF < 50%, as well as LVEF >= 40% and LVEF < 40%, we corrected Table 3 and Supplementary Table S2. The corrected values now ensure the accuracy of our diagnostic performance metrics. 

2. The omission of ARNi from the model strikes me as strange. Of the drugs that could be included in the RASi-variable (ACEi, ARB, and ARNi), ARNi is likely the far best at distinguishing between the EF phenotypes. Moreover, ARNi technically includes an ARB. How much this omission affects model performance is difficult to speculate without knowing the time period of enrollment and how many patients used ARNi. I highly suggest to consider including it in the RASi variable, as well as to disclose ACEi/ARB and ARNi separately in the characteristics table. 

Reply: Thank you for your insightful feedback regarding the inclusion of ARNi in the RAAS agents variable. Our decision of not including them was based on the fact that the original algorithm developed in the Swedish Heart Failure Registry did not include ARNi due to the timeframe of the study and the adoption of these medications during that period. Specifically, the first prescriptions of ARNi in Sweden began in April 2016, and the study from A. Uijl included patients between 2000 and 2012.

In light of your comment, we have additionally checked our dataset from the Cardiology Centers of the Netherlands about the use of ARNi, but we did not find any mention of it. Upon consultation with the cardiologists at CCN, it was confirmed that regular prescription of ARNi, began only around 2021, while our analysis includes participants from 2007 to 2018. 

Therefore, because our study period predates the widespread use of ARNi, incorporating them in the RAAS agents variable is unfortunately not feasible due to the lack of data. 

Finally, we would like to point out that ARNi is now being recognized for its benefits across all HF phenotypes, including HFpEF and HFmrEF, as demonstrated in the 2023 PARAGLIDE trial. We understand the potential impact of ARNi and will consider this in future analyses, as more recent data becomes available. 

3. Please clarify in the methods between which dates these patients had their index dates. It says CCN data were accessed between Feb 2022 and Jan 2024, but I interpret this literally as data access and not when the patients presented in the clinic.

Reply: we agree with the reviewer that this is an important information and should be clearly stated in the manuscript. Our analysis includes patients who were diagnosed with HF between June 2007 and February 2018. We have added this information in the methods section on page 4. 

4. Page 11: "While we acknowledge that the dosage of these medications may be subject to adjustments over time as part of HF management, it is unlikely that the discontinuation or initiation of these drugs would occur significantly long after the initial diagnosis of HF. This is because these HF medications represent the cornerstone of HF treatment and are typically prescribed and monitored closely in clinical practice." This sentence seems at odds with observations from several health care systems. E.g. in EVOLUTION-HF (10.1016/j.jchf.2022.08.009) 33-38% of patients discontinued ACEi/ARB within 12 months.

Reply: We appreciate your observation regarding the discontinuation rates of ACEi/ARB in HF patients, as highlighted in the EVOLUTION-HF study. We recognize that our statement may not align with these observations. Accordingly, we have revised this paragraph as follows, and included a reference to the EVOLUTION-HF study (page 11). 

“We acknowledge that the dosage of these medications may be subject to adjustments over time as part of HF management, and discontinuation and initiation of these drugs might as well occur at notable rates. For instance, the EVOLUTION-HF study reported that 33-38% of patients discontinued ACE-inhibitors and angiotensin receptor blockers within 12 months [16]. However, these medications still represent the cornerstone of HF treatment and are typically prescribed and monitored closely in clinical practice, especially in the initial stages following diagnosis.”

5. I suggest to change the term "mid-range" EF to "mildly reduced" EF to align with currently accepted terminology.

Reply: we have revised this accordingly. 

6. Conclusion: Just a minor comment but was the distinction between primary care and GP settings made on purpose? "[...] where direct measurements of EF are not routinely available, such as in primary care, or other non-cardiology health settings such as GP settings."

Reply: We acknowledge that the terms "primary care" and "GP settings" can be used interchangeably, and the distinction we made was not intentional. Both terms refer to the settings where general practitioners provide initial and ongoing patient care. We have revised the conclusion to use consistent terminology to avoid confusion: "…where direct measurements of EF are not routinely available, such as in primary care settings."

---

## [Decision Letter · Decision Letter 2]

23 Aug 2024

Simplification of a Registry-Based Algorithm for Ejection Fraction Prediction in Heart Failure Patients: Applicability in Cardiology Centres of the Netherlands

PONE-D-24-04912R2

Dear Dr. Canto,

We’re pleased to inform you that your manuscript has been judged scientifically suitable for publication and will be formally accepted for publication once it meets all outstanding technical requirements.

Kind regards,

Satoshi Higuchi

Academic Editor

PLOS ONE

Reviewers' comments:

Reviewer's Responses to Questions

**Comments to the Author**

1. If the authors have adequately addressed your comments raised in a previous round of review and you feel that this manuscript is now acceptable for publication, you may indicate that here to bypass the “Comments to the Author” section, enter your conflict of interest statement in the “Confidential to Editor” section, and submit your "Accept" recommendation.

Reviewer #2: All comments have been addressed

2. Is the manuscript technically sound, and do the data support the conclusions?

Reviewer #2: Yes

3. Has the statistical analysis been performed appropriately and rigorously? 

Reviewer #2: Yes

4. Have the authors made all data underlying the findings in their manuscript fully available?

Reviewer #2: No

5. Is the manuscript presented in an intelligible fashion and written in standard English?

Reviewer #2: Yes

6. Review Comments to the Author

Reviewer #2: I thank the authors for thoroughly addressing my previous comments. I have no further remarks to be considered.

7. PLOS authors have the option to publish the peer review history of their article (what does this mean?). If published, this will include your full peer review and any attached files.

Reviewer #2: No

---

## [Editor Report · Acceptance letter]

4 Sep 2024

PONE-D-24-04912R2 

PLOS ONE

Dear Dr. Canto, 

I'm pleased to inform you that your manuscript has been deemed suitable for publication in PLOS ONE. Congratulations! Your manuscript is now being handed over to our production team.

Kind regards, 

on behalf of

Dr. Satoshi Higuchi 

Academic Editor

PLOS ONE